# Topical TMPRSS2 inhibition prevents SARS-CoV-2 infection in differentiated human airway cultures

Wenrui Guo[1],*, Linsey M Porter[1],*, Thomas WM Crozier[2],*, Matthew Coates[1], Akhilesh Jha[1], Mikel McKie[3], James A Nathan[2] , Paul J Lehner[2], Edward JD Greenwood[2] , Frank McCaughan[1]

**Background:** There are limited effective prophylactic/early treatments for severe acute respiratory syndrome coronavirus 2 (SARS-CoV-2) infection. Viral entry requires spike protein binding to the angiotensin-converting enzyme-2 receptor and cleavage by transmembrane serine protease 2 (TMPRSS2), a cell surface serine protease. Targeting of TMPRSS2 by either androgen blockade or direct inhibition is in clinical trials in early SARS-CoV-2 infection. **Methods:** We used differentiated primary human airway epithelial cells at the air–liquid interface to test the impact of targeting TMPRSS2 on the prevention of SARS-CoV-2 infection. **Results:** We first modelled the systemic delivery of compounds. Enzalutamide, an oral androgen receptor antagonist, had no impact on SARS-CoV-2 infection. By contrast, camostat mesylate, an orally available serine protease inhibitor, blocked SARS-CoV-2 entry. However, oral camostat is rapidly metabolised in the circulation, with poor airway bioavailability. We therefore modelled local airway administration by applying camostat to the apical surface of differentiated airway cultures. We demonstrated that a brief exposure to topical camostat effectively restricts SARS-CoV-2 infection. **Conclusion:** These experiments demonstrate a potential therapeutic role for topical camostat for pre- or post-exposure prophylaxis of SARS-CoV-2, which can now be evaluated in a clinical trial.

## Introduction

A novel subtype of coronavirus, severe acute respiratory syndrome coronavirus 2 (SARS-CoV-2) was first reported in December 2019 and has led to a global pandemic. There is compelling in vitro and in vivo evidence that angiotensin-converting enzyme-2 (ACE2) and transmembrane serine protease 2 (TMPRSS2) are required for SARS-CoV-2 entry (Breining et al, 2021; Hoffmann et al, 2020b). SARS-CoV-2 binds ACE2 at the cell surface through its spike protein. TMPRSS2 is a serine protease that cleaves the spike protein thus priming membrane fusion and cellular entry (Heurich et al, 2014).

A critical question is whether ACE2 and TMPRSS2 are suitable targets for the prevention or treatment of SARS-CoV-2. Any useful intervention targeting viral entry would need to be administered at the early stage of viral infection to target viral replication rather than later phase COVID-19 disease which is thought to be immune-mediated. The main entry point for SARS-CoV-2 infection are the TMPRSS2 and ACE2 expressing ciliated epithelial cells within the nasal cavity and bronchial airways (Ruiz García et al, 2019; Martines et al, 2020; Sungnak et al, 2020 Preprint). It follows that for preventative therapies, the use of differentiated airway cells is critical to modelling therapeutic efficacy in vitro.

Prior work suggests that soluble ACE2 could act as a decoy receptor for spike protein and reduce SARS-CoV-2 viral load (Chan et al, 2020), and clinical trials testing this strategy are underway (Zoufaly et al, 2020). An alternative approach is to modulate the cleavage of spike protein by inhibiting TMPRSS2 activity, by (1) using androgen receptor (AR) antagonists to suppress androgen-sensitive TMPRSS2 expression (Samuel et al, 2020; Ziegler et al, 2020), or direct inhibition of the serine protease activity of TMPRSS2 (Hoffmann et al, 2020b; Stopsack et al, 2020).

TMPRSS2 expression is regulated by androgen signalling via AR binding at 13 and 60 kb upstream of the TMPRSS2 transcriptional start site (Wang et al, 2007). Translocation of this AR binding site is both common and pathogenic in androgen-dependent prostate cancer (Rahim & Uren, 2013). Androgen signalling may underpin the epidemiological evidence linking male gender with worse clinical outcomes in COVID-19 (Pradhan & Olsson, 2020). Furthermore, a retrospective survey of cancer patients (Chakravarty et al, 2020; Montopoli et al, 2020) reported a potential impact of androgen deprivation therapy on COVID-19 severity. As a result, clinical trials of short-course androgen deprivation in early, non-hospitalised male COVID-19 patients have started recruitment (ClinicalTrials.gov Identifier: NCT04446429; ClinicalTrials.gov Identifier: NCT04475601).

Camostat mesylate, a serine protease inhibitor, directly inhibits TMPRSS2 (Shirato et al, 2013). Experiments mimicking the systemic

[1]Department of Medicine, Addenbrookes Hospital, University of Cambridge, Cambridge, UK   [2]Department of Medicine, Cambridge Institute of Therapeutic Immunology and Infectious Disease, University of Cambridge, Cambridge, UK   [3]Medical Research Council, Biostatistic Unit, Cambridge, UK

Correspondence: ejdg2@cam.ac.uk; fm319@cam.ac.uk
*Wenrui Guo, Linsey M Porter, and Thomas WM Crozier contributed equally to this work.

delivery of camostat have shown that it is effective in reducing viral entry. Camostat reduced pseudotyped SARS-CoV-2 viral entry in differentiated airway epithelial cells (Hoffmann et al, 2020b) and reduced the already modest levels of wild type viral entry in distal airway organoids (Samuel et al, 2020; Youk et al, 2020). A potential benefit of camostat is that it is orally delivered as well as being well-tolerated and inexpensive (Breining et al, 2020). It has been licensed in Japan since 1985 and is in regular use in Japan and South Korea for patients with chronic pancreatitis. Several clinical trials in patients with COVID-19 have now been undertaken (ClinicalTrials.gov Identifier: NCT04455815; ClinicalTrials.gov Identifier: NCT04608266) (Gunst et al, 2021).

Given the potential therapeutic role for targeting TMPRSS2 in SARS-CoV-2 infection, we assessed TMPRSS2 regulation by androgens in primary airway cells and how androgen antagonism or camostat treatment affected SARS-CoV-2 infection in this clinically relevant system. We show that direct inhibition of TMPRSS2 with camostat is effective in reducing SARS-CoV-2 infection, but similar findings were not observed with androgen antagonism. Because camostat is rapidly metabolised in the circulation, we predicted it would have poor airway bioavailability when administered orally. While this manuscript was in revision, clinical trials have confirmed the lack of clinical impact of the oral preparation. We therefore tested whether topically delivered camostat would be effective. This mimics the local delivery of camostat to the airway. We were aware that the nasal transit time is estimated at 20 min (Englender et al, 1990; Plaza Valía et al, 2008) and therefore limited the topical exposure to periods of 15 min only. Our results show that brief exposure to topical camostat in primary human airway cells at the air–liquid interface (ALI) effectively reduces SARS-CoV-2 infection, providing a potentially safe and effective means to deliver TMPRSS2 inhibition to the site of viral entry.

# Results

### ACE2/TMPRSS2 expression in differentiated human airway cells in response to DHT and enzalutamide

Ciliated airway epithelial cells are considered the main initial cell of entry for SARS-CoV-2, although goblet and secretory (Club) cells can also be infected. To model viral entry in vitro we used differentiated human bronchial epithelial cells (hBECs) at the ALI. Submerged primary hBECs have a basal cell phenotype; they are effectively stem cells of the airway epithelium and are not ciliated (Crystal et al, 2008; Walters et al, 2013). When cultured at ALI in an appropriate medium, hBECs differentiate to a pseudostratified epithelium that comprise basal, secretory, goblet, and ciliated cells (Sachs et al, 2003) (Figs 1A and B and S1). The expression of ACE2 and TMPRSS2 increased significantly in differentiated cells cultured at ALI compared with submerged standard culture (Fig 1C). This has previously been described for ACE2 (Jia et al, 2005), but these data show that TMPRSS2 is also differentially regulated when airways cells are cultured exposed to air. Given the central importance of both proteins in SARS-CoV-2 infection, this reinforces the importance of differentiated ALI cultures in studying the initial impact of viral infection.

We next investigated the impact of testosterone signalling on TMPRSS2 and ACE2 expression in our ALI model. Differentiated hBECs (Donor B1, male) were established at the ALI and cultured for

2 d in the absence of androgen stimulation before 24-h exposure to 5-dihydroxytestosterone (DHT) and/or enzalutamide (Fig 1A). DHT is the 5-$\alpha$-reductase metabolite of testosterone and is a more potent agonist of the AR than testosterone. Enzalutamide is an androgen signalling antagonist. Both DHT and enzalutamide had the anticipated impact on canonical AR genes (FKBP5 and NDRG1) in an androgen-sensitive prostate cancer cell line (Fig S2) and in differentiated hBECs (Fig 1D, Donor B1).

On treatment of Donor B1 cells with DHT there was a modest, but not significant, induction of TMPRSS2. Treatment with enzalutamide alone had no impact on TMPRSS2 mRNA expression, although it reverted any impact of DHT to baseline. Enzalutamide, in combination with DHT, also caused a modest reduction of ACE2 mRNA expression (Fig 1E).

### AR antagonism does not significantly impact on SARS-CoV-2 infection in human airway epithelial cells (hAECs)

Next, we infected differentiated hBECs (Donor B1) with SARS-CoV-2, with or without prior treatment with DHT and/or enzalutamide. These compounds were added to the basal chamber of the Transwell mimicking their systemic delivery. 72 h post inoculation, infection was assessed using a quantitative flow cytometric assay for cells expressing nucleoprotein protein, as a measure of viable infected cells (Fig 1F). The well-to-well variation in infection in primary cells has been reported by multiple groups (Purkayastha et al, 2020; Hoffmann et al, 2020b; Mulay et al, 2021). We found no significant impact of either DHT or enzalutamide on the proportion of cells infected in this assay (Fig 1F).

### Basal camostat is effective at blocking SARS-CoV-2 infection

Although manipulating AR signalling was ineffective in modulating SARS-CoV-2 infection, we considered that direct inhibition of TMPRSS2 may have more profound effects. We therefore first treated differentiated hBECs at ALI with camostat for 48 h in the basal chamber (Fig 2A), as a surrogate for the potential impact of systemic therapy. This was a proof-of-principle experiment in our system—we were aware that the maximal reported concentration of the primary camostat metabolite is <1 $\mu$M (Breining et al, 2021) and so oral camostat was an implausible therapeutic strategy. However, basal camostat did significantly reduce the number of infected cells with or without DHT pretreatment (Fig 2B). This was consistent with multiple other reports (Suzuki et al, 2020 Preprint; Hoffmann et al, 2020a Preprint).

Parallel experiments using immunofluorescence to detect the S2 subunit of the viral spike protein corroborated the flow cytometry data (Fig 2C). Cells were fixed 72 h after infection. Again, consistent with previous reports, spike protein colocalised with ACE2 expression. Pretreatment with camostat markedly reduced the proportion of cells expressing spike protein at 72 h (Fig 2D).

Histological analysis of hBECs following 48 h basal camostat treatment showed no toxicity (Fig 2E), and there was no excess release of lactate dehydrogenase (LDH) in response to camostat in ALI cultures derived from bronchial or nasal cells (Fig 2F). However, given the rapid metabolism of camostat in the systemic circulation (half-life <1 min) (Breining et al, 2021) we were aware that the airway bioavailability is likely to be extremely low. This is consistent with the negative results now reported in the trials of oral camostat (Gunst et al, 2021;

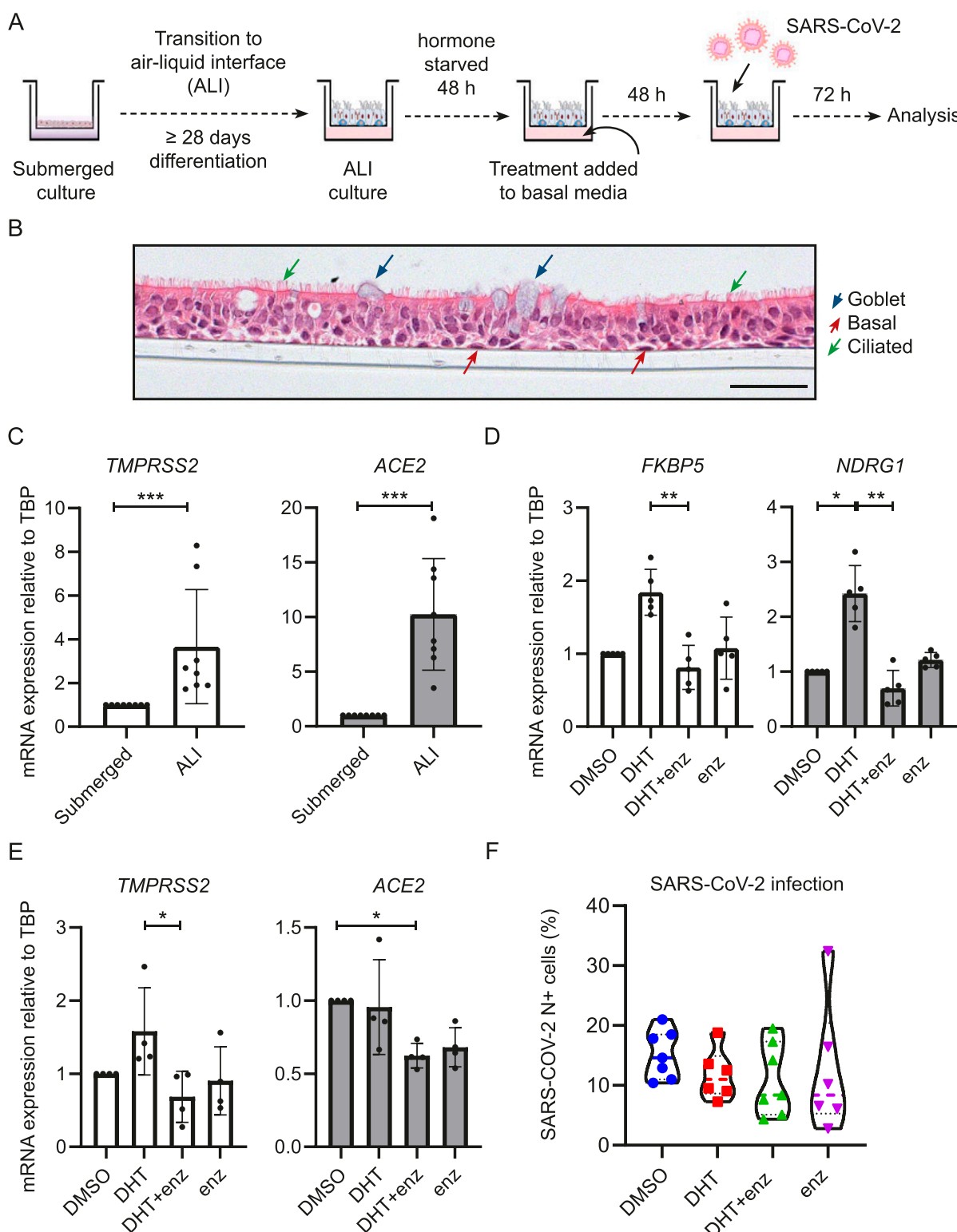

**Figure 1.   Modulation of androgen signalling does not impair severe acute respiratory syndrome coronavirus 2 infection of air–liquid interface (ALI) differentiated human bronchial epithelial cells (hBECs).**

Scale bar: 50 µm. **(A)** Schematic representation of the hBEC ALI experimental setup. **(B)** H&E staining of fully differentiated pseudostratified epithelium hBECs at the ALI. Ciliated cells, goblet cells, and basal cells are indicated. H&E image was taken from an ALI culture of cells from Donor B1. **(C)** Transmembrane serine protease 2 (TMPRSS2) and angiotensin-converting enzyme-2 (ACE2) expression (mRNA) both increase on culture at the ALI compared to submerged cell culture. P-values were calculated using an unpaired Mann–Whitney U test. P = 0.0002 and 0.0002 for TMPRSS2 and ACE2 on ALI compared with submerged, respectively. Each dot represents the mean of

Ono Pharmaceuticals, 2021). We therefore hypothesised that topically administered camostat, mimicking nasal or inhaled/nebulised camostat, may prevent SARS-CoV-2 infection.

### Topical camostat retains efficacy in preventing SARS-CoV-2 infection

We mimicked local therapy by applying topical 50 $\mu$M camostat to the air-exposed side of differentiated hBECs (Donor B1) at ALI for either two "doses" separated by 24 h or a single "dose" ~4 h before infection (Fig 3A). Each topical dose comprised application of camostat in solution to the apical surface for 15 min before removal by aspiration. 15 min was chosen because the nasal transit time in humans is ~20 min so that 15-min exposure is an accurate model of topical therapeutics in vivo (Englender et al, 1990; Plaza Valía et al, 2008). Cells were then infected with SARS-CoV-2 and infection analysed at 72 h by flow cytometry (Fig 3B). The proportion of cells infected with SARS-CoV-2 showed a clear dose-dependent reduction. Consistent with the flow cytometry data, immunofluorescence analysis showed that topical treatment with camostat markedly reduced spike protein expression (Fig S3A). Cross-sections of the epithelium stained using nucleoprotein antibody (Fig 3C) corroborated this result. Apical camostat treatment was also effective at reducing wild type SARS-CoV-2 infection in nasal ALI cultures from a different individual–Donor N3 (Fig S3B). We concluded that two short exposures to topical camostat were effective in preventing SARS-CoV-2 infection.

We then performed a series of experiments to test whether topical camostat was effective in preventing infection by the SARS-CoV-2 B.1.1.7 variant. Bronchial epithelial cells from two donors were infected after pretreatment with SARS-CoV-2. Quantitative immunofluorescence of viral nucleoprotein imaging of whole Transwell scans was used to assess the percentage of cells infected (Fig 3D). There was an attenuated impact of 50 $\mu$M with the B.1.1.7 variant compared to the earlier experiments, but 300 $\mu$M was sufficient to effectively block infection (Fig 3E). Similar results were obtained in replicate experiments and using primary nasal ALI cultures from three donors (Figs 3E and S4). Furthermore, topical doses of camostat up to 1 mM did not cause any gross histological toxicity (Fig S5). Sustained treatment with basal camostat at both 50 and 300 $\mu$M concentrations was sufficient to prevent infection (Fig S6), but as noted above, these concentrations are not achievable in vivo.

Finally, we undertook a series of post-infection intervention experiments in which we treated with camostat after viral infection either supplementing basal media (at 24 h and refreshed at 48 h) and topical (two 15 min doses at 24 and 48 h) (Fig 4A). As expected, the sustained basal treatment was more effective that the topical treatment but as noted before the pharmacokinetics means these concentrations are unachievable. Topical camostat, was sufficient to significantly reduce cellular infection for all donors (B1 and N3-5) (Fig 4B–D). Therefore, topical TMPRSS2 inhibition when delivered in a clinically relevant and achievable dose to differentiated airways cells markedly restricts SARC-CoV-2 cellular infection.

## Discussion

### AR antagonism is unlikely to be effective at limiting viral entry to airway cells

We have used in vitro systems to model the likely impact of androgen antagonism as a means of preventing early SARS-CoV-2 infection (Bennani & Bennani-Baiti, 2020; Samuel et al, 2020). Our experiments suggest that enzalutamide is unlikely to be useful for preventing TMPRSS2-mediated cleavage and subsequent SARS-CoV-2 infection in cells of the conducting airways. A recent commentary questioned the potential use of enzalutamide to prevent COVID-19 (O'Callaghan et al, 2020). The authors calculated that 434 individuals would need to be treated with enzalutamide to prevent a single case of COVID-19 and noted the unpleasant side effect profile of enzalutamide (O'Callaghan et al, 2020). Our data are in contrast to a recent report suggesting anti-androgens (5-$\alpha$ reductase inhibitors) may reduce SARS-CoV-2 infection in alveolar organoids (Samuel et al, 2020). However, there is a respiratory epithelial infectivity gradient with proximal airway cells significantly more susceptible than the distal airway (Hou et al, 2020), and our quantitative assays of airway infection suggest that compared with camostat mesylate, androgen antagonism is ineffective at preventing proximal airway infection.

### Topical camostat restricts SARS-CoV-2 infection

The pharmacokinetics of systemic camostat render the standard oral preparation unlikely to be effective outside the gastrointestinal tract. Camostat mesylate is almost instantaneously metabolised in the circulation to an intermediate metabolite with significantly less activity (Breining et al, 2020; Hoffmann et al, 2020a Preprint). Although well tolerated at high oral doses, the maximum concentration of the metabolite attained in the circulation at standard dosing is 0.18 $\mu$M (Bittmann et al, 2020), significantly lower than the sustained "basal" concentrations used in this and other studies to prevent

---

three technical replicates from one individual experiment. All replicates were performed using hBECs from Donor B1. **(D)** mRNA expression of canonical androgen receptor canonical genes, FKBP5 and NDRG1 after 48 h treatment with 10 nM DHT with or without 10 $\mu$M enzalutamide. *P*-values were calculated using a Kruskal–Wallis test with Dunn's correction for multiple comparisons (between all conditions). $P = 0.0061$ for *FKBP5* in HBEC-ALI on DHT + enz arm versus DHT; $P = 0.0183$ and $P = 0.0015$ for *NDRG1* in HBEC-ALI on DHT arm versus DMSO and DHT arm versus DHT + enz arm, respectively. Each dot represents the mean of three technical replicates from one individual experiment. All replicates were performed using cells from Donor B1. **(E)** mRNA expression of ACE2 and TMPRSS2 in HBEC-ALI after 48-h treatment with 10 nM DHT with or without 10 $\mu$M enzalutamide. *P*-values were calculated using a Kruskal-Wallis test with Dunn's correction for multiple comparisons (between all conditions). $P = 0.0212$ for TMPRSS2 on DHT + enz arm versus DHT; $P = 0.0268$ for ACE2 on DHT + enz arm versus DMSO. Each dot represents the mean of three technical replicates from one individual experiment. All replicates were performed using cells from Donor B1. **(F)** Violin plots showing the quantification of severe acute respiratory syndrome coronavirus 2 infected total cells post treating with 10 nM DHT, 10 $\mu$M enzalutamide along or in the combination with DHT. All compounds were added to the basal chamber. Each symbol represents one Transwell from two independent experiments. *P*-values were calculated using a Kruskal–Wallis test with Dunn's correction for multiple comparisons (between all conditions). All *P*-values > 0.5. All replicates were performed using cells from Donor B1.

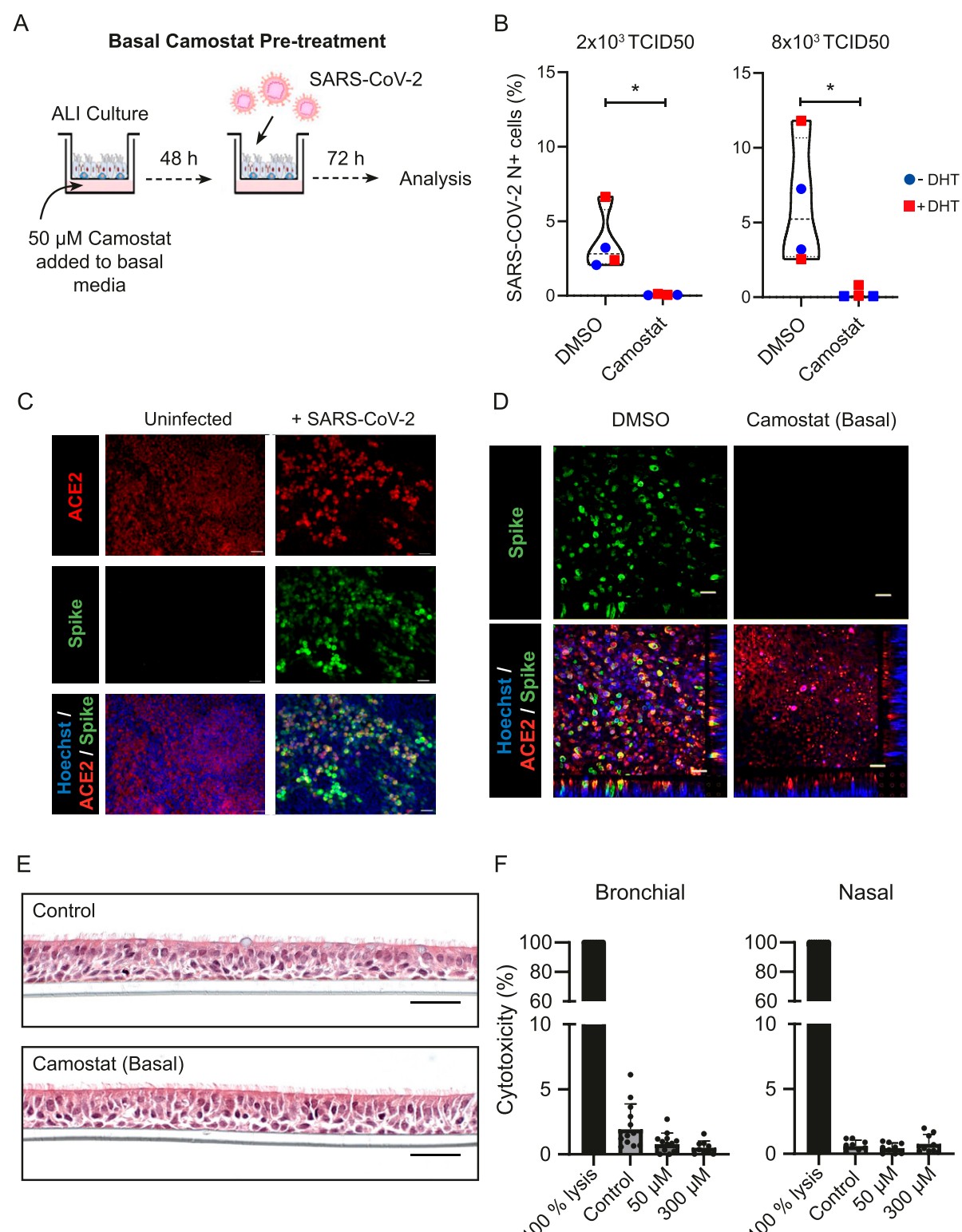

**Figure 2. Basal Camostat blocks severe acute respiratory syndrome coronavirus 2 (SARS-CoV-2) infection.**
**(A)** Schematic of experiment. Camostat was added to the basal media 48 h before SARS-CoV-2 infection. **(B)** Violin plots showing the quantification of SARS-CoV-2 infected total cells after treatment with 50 $\mu$M Camostat in the basal chamber with or without DHT. Each dot represents a Transwell. Experiment was performed twice with wells exposed to either $2 \times 10^3$ or $8 \times 10^3$ TCID50 SARS-CoV-2 preparations in 50 $\mu$l. There was a significant reduction in viral infection in the wells treated with camostat. Samples also treated with DHT are shown in red. P-values were calculated using a Mann–Whitney U test, P = 0.0286 at $2 \times 10^3$ TCID50 and P = 0.0286 at $8 \times 10^3$ TCID50. All replicates were performed using cells from Donor B1. **(C)** Representative immunofluorescent image of human bronchial epithelial cell–air–liquid interface (ALI) after SARS-CoV-2 infection

SARS-CoV-2 infection (Bittmann et al, 2020; Breining et al, 2020; Suzuki et al, 2020 Preprint; Youk et al, 2020; Hoffmann et al, 2020a Preprint, 2020b). These experiments provide a proof-of-principle for TMPRSS2 inhibition being a rational and potentially effective strategy but to successfully translate to clinical practice a formulation that avoids the rapid metabolism in the circulation is critical.

Our findings show that early administration of a topical formulation at clinically achievable doses may also be very effective in reducing SARS-CoV-2 infection. Local airway delivery of camostat was previously proposed as an inhibitor of a serine protease that regulates sodium channel flux (Coote et al, 2009; Rowe et al, 2013). The objective was to change mucus characteristics in cystic fibrosis. Although TMPRSS2 is the key protease targeted by camostat in SARS-CoV-2 inhibition, it is not possible to formally exclude a role for inhibition of other serine proteases in the observed results.

Coote and colleagues Coote et al (2009) reported preclinical data in hBECs at ALI, in which cells were exposed to up to 30 $\mu$M camostat topically for 90 min. Camostat inhibition of sodium channel flux persisted at least 6 h after treatment, despite multiple washes with warmed PBS, demonstrating that topical camostat can have a sustained impact. This duration is consistent with the proposed "pseudo-irreversible" mechanism of serine protease inhibition by camostat mediated by covalent binding to the target (Breining et al, 2020) and is consistent with the apparent sustained inhibition in our system despite the administration being for 15 min periods only.

A number of in vivo preclinical experiments using airway formulations of camostat have also been reported, including the administration of nebulised camostat to anaesthetised sheep (Coote et al, 2009). Up to 60 mg camostat was delivered by nebulisation in 3 ml, equating to a maximal concentration of 50.2 mM, markedly higher than we used, without toxicity. Importantly, the duration of activity using a surrogate of therapeutic activity in vivo, was at least 5 h, again consistent with the proposed model of covalent binding to target serine proteases. Further unpublished studies in dogs suggested a mild and reversible inflammatory response in individual animals (lung parenchyma) to inhaled camostat (Novartis, personal communication); but no significant toxicity associated with high-dose nasal administration. Therefore, nebulised (or inhaled) camostat to lower airways was deemed to not be a safe route of administration and further development of this route of delivery terminated. The nasal route, however, appeared safe.

Therefore, a subsequent early phase trial focussed on the nasal route - topical camostat was delivered to the nose in volunteers with cystic fibrosis (Rowe et al, 2013). The doses tested were in the range 5–1,600 $\mu$g and nasal potential difference was used as a surrogate of target engagement. The authors reported target engagement and therapeutic efficacy with a 50% maximal effect dose estimated at 18 $\mu$g/ml, equivalent to 45.2 $\mu$M. Although three serious

adverse events were recorded, none were judged to be because of the study medication. Furthermore, the other adverse events were minor and not reported in those administered 200 $\mu$g or less. Camostat was not detected in the systemic circulation after nasal dosing. The primary metabolite was detected at 3 h, only in those individuals given the highest dose of 1,600 $\mu$g. Assuming a per nostril epithelial lining fluid volume of 800 $\mu$l (Kaulbach et al, 1993), the effective concentration after delivery of 1,600 $\mu$g is 4.0 mM, substantially higher than the doses shown to be effective in this study against two SARS-CoV-2 variants.

Importantly, a 200 $\mu$g dose would be in excess of 400 $\mu$M, with the data in Fig 3 suggesting 300 $\mu$M would have a greater than 90% inhibitory effect on cellular infection. Together, these data suggest that safe and sustained nasal serine protease inhibition can be achieved in primary hAECs using a nasal camostat spray.

TMPRSS2 is a potentially important target for repurposed medications and is an attractive strategy for prophylaxis as the target is the host cell rather than SARS-CoV-2. Both direct inhibition and androgen deprivation have been suggested as strategies to reduce TMPRSS2 activity. Our data do not support the use of AR antagonists but suggest that direct inhibition of TMPRSS2 will have activity against multiple SARS-CoV-2 variants including B.1.1.7, recently shown to have increased transmissibility (Davies et al, 2021). Consistent with our observations, Hoffmann and colleagues Hoffmann et al (2021) have also shown that camostat effectively inhibits variant cell entry in the Caco-2 colorectal carcinoma cell line. They demonstrated this using pseudotyped viral particles engineered to express a series of key variants (B.1.1.7, B1.351, and P.1) implicated as variants that may alter virus-host cell interactions and confer resistance to antibodies (Hoffmann et al, 2021).

The impact of an oral camostat preparation in COVID-19 has now been reported and the results were negative (Gunst et al, 2021; Ono Pharmaceuticals, 2021). Our data provide a rationale for testing camostat as a local airway administered prophylaxis or early treatment for SARS-CoV-2 infection. When effective, local delivery has the advantages of reducing the systemic dose and associated side effects while delivering the drug to the site of disease. Camostat is well tolerated in the upper airway, a plausibly effective dose was readily achieved and the compound is soluble in saline and cheap to manufacture (Coote et al, 2009; Breining et al, 2020).

While this paper was in review, studies have been published suggesting that nasal TMPRSS2 inhibition using nafamostat is effective in preventing SARS-Co-V2 infection in small animal models (Li et al, 2021). These data are complementary to our study and strongly support the strategy being moved into a clinical trial. Nafamostat is a more potent inhibitor in cell-free biochemical assays and cellular assays but, unlike camostat, does not yet have Phase 1 safety data and prior United States Food and Drug Administration (FDA) approval for

---

using antibodies to angiotensin-converting enzyme-2 and the S2 subunit of spike protein. Uninfected human bronchial epithelial cell-ALI was used to verify specificity. Scale bar: 100 $\mu$m. This image used cells from Donor B1. **(D)** Impact of 48 h of 50 $\mu$M basal camostat on cellular infection. Representative immunofluorescent images of human airway epithelial cell-ALI post SARS-CoV-2 infection using antibodies to angiotensin-converting enzyme-2 and the S2 subunit of spike protein. Scale bar: 100 $\mu$m. This image used cells from Donor B1. **(E)** H&E staining showing normal morphology in cells exposed to 50 $\mu$M camostat for 48 h. This image used cells from Donor B1. **(F)** LDH release assay as a measure of cytotoxicity after 48-h treatment using basal camostat on ALI cultures derived from bronchial or nasal cells. Complete lysis is shown as a positive control. Doses of up to 300 $\mu$M (basal) did not cause increased cytotoxicity. Each dot represents a Transwell and for each condition three biological replicates were performed including at least three technical replicates. These experiments were performed using cells from Donors B1 and N3. P-values were calculated using a Kruskal–Wallis test with Dunn's correction for multiple comparisons (between all conditions). There was no significant difference between 50 and 300 $\mu$M camostat treatment and the control wells.

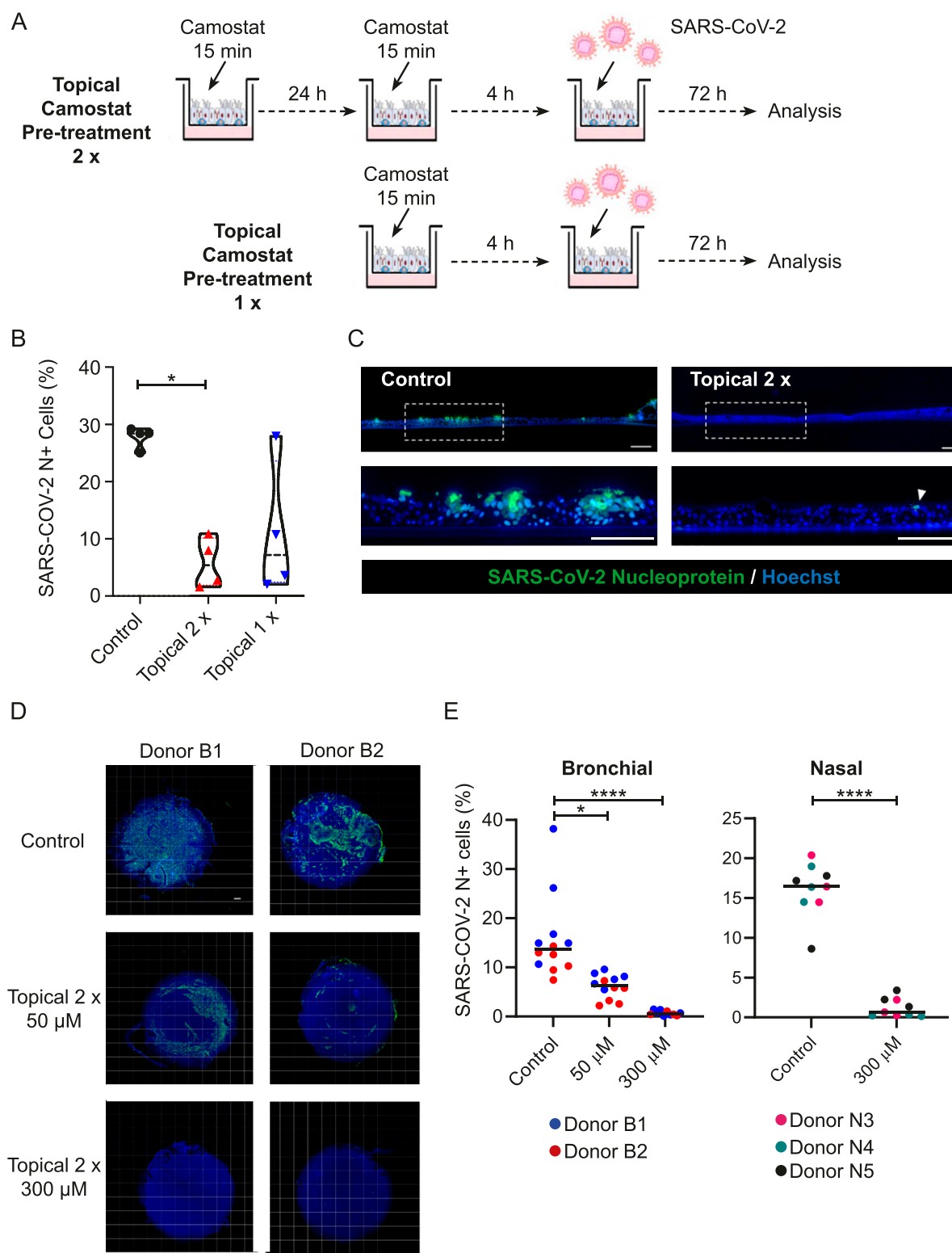

**Figure 3. Camostat applied to the apical surface blocks severe acute respiratory syndrome coronavirus 2 (SARS-CoV-2) infection.**
**(A)** Schematic of experimental design for topical camostat treatment before SARS-CoV-2 infection. **(B)** Violin plots showing the quantification of infected cells when pretreated with camostat in the apical chamber. *P*-values were calculated using a Kruskal–Wallis test with Dunn's correction for multiple comparisons (apical treatment conditions compared with control). *P* = 0.0285 and *P* = 0.0997 for topical 2× and topical 1× versus control, respectively. Each dot represents a Transwell. Experiments were performed using cells from Donors B1 and B2. **(C)** Cross-sectional images of IF staining for nucleoprotein on formalin-fixed paraffin-embedded air–liquid interface cultures. Scale bars: 100 *μ*m. **(D)** Air–liquid interface cultures were pre-treated with 2 × 15 min of topical camostat at the indicated dose before infection with 1 × 10⁴ TCID50

use in humans as a nasal preparation. Our study also confirms the potential for ex vivo differentiated human primary airway model systems to reduce and replace animal experiments in the context of therapeutic repurposing/reformulation studies where compound safety has already been established.

The recent success of the SARS-CoV-2 vaccine trials in protecting against COVID-19 disease has had a major impact on the pandemic (Polack et al, 2020; Baden et al, 2021; Voysey et al, 2021). Furthermore, there is now evidence that monoclonal antibodies delivered subcutaneously or intravenously are effective in the pre-hospital setting in both the prevention of symptomatic COVID-19 and treatment of early phase infection (Chen et al, 2021; Weinreich et al, 2021).

Despite these critical advances, there remains an urgent unmet need for a safe, cheap, easily administered, effective therapeutic with potential efficacy in pre- or post-exposure prophylaxis, and in the reduction of transmission, or progression, in asymptomatic/early SARS-CoV-2 infection. Our work suggests that topical delivery of camostat mesylate should now be clinically evaluated for these indications.

# Materials and Methods

### Primary hAEC: ALI model

hBECs were purchased from Lonza (Donor B1—Bronchial 1) or were expanded directly from either bronchial brushings from a main airway at bronchoscopy (Donor B2 – Bronchial 2) or a nasal brushing from the inferior turbinate from patients at Cambridge University Hospitals National Health Service Trust (Research Ethics Committee Reference 19/SW/0152) (Donors N3-N5 – Nasal 3–5). In brief, primary airway cells at passage 2 were expanded in PneumaCult-Ex Plus Medium (Cat. no. #05040; Stemcell) then seeded on collagen (Cat. no. #354236; Corning) coated 24-well Transwell inserts with 0.4-$\mu$m pores (Cat. no. #353095; Falcon) until fully confluent. Once confluent, the cells were taken to the ALI and cultured with PneumaCult-ALI Medium (Cat. no. #05021; Stemcell) for at least 28 d before conducting experiments. On the day of harvest, adherent cells were washed once with PBS and incubated in Accutase (Cat. no. #07920; Stemcell) at 37°C for 15 min. Cells were dissociated by gentle pipetting and then neutralised with DMEM/10% FBS. Whole cell pellets were collected by centrifuge at 300$g$ for 5 min at room temperature. Specific cells used were hBECs derived from a non-smoking donor (Cat. no. # CC-2540; Lonza, male, Donor B1); hBECs from a male smoking donor undergoing bronchoscopy for a non-cancer indication (Donor B2), and nasal epithelial cells from three female patients (Donors N3, N4, and N5).

Test compounds: 5-dihydroxytestosterone (DHT) (Cat. no. S4757; Selleckchem), enzalutamide (Cat. no. S1250; Selleckchem), camostat mesylate (Cat. no. 3193; Tocris) were added to the basal chamber of

the Transwell as indicated and dissolved in DMSO (max final concentration 0.1%) (enzalutamide) or PBS (camostat). Differentiated cells were cultured for 2 d in the absence of androgen stimulation before exposure to DHT or enzalutamide for 24 h. For combination treatments 24-h enzalutamide pretreatment was applied before DHT exposure.

For topical delivery of camostat mesylate, the apical chamber was washed with PBS before 50 $\mu$l of the indicated concentration of camostat in PBS (or PBS alone in control wells) was added to the apical chamber for 15 min before being aspirated.

### SARS-CoV-2 infection assays

The SARS-CoV-2 viruses used in this study are the clinical isolates named "SARS-CoV-2/human/Liverpool/REMRQ0001/2020" (Chu et al, 2020; Patterson et al, 2020) and "SARS-CoV-2 England/ATACCC 174/2020" (Lineage B.1.1.7). Stocks were sequenced before use and the consensus matched the expected sequence exactly. Viral titre was determined by 50% tissue culture infectious dose (TCID$_{50}$) in Huh7-ACE2 cells.

For viral infection, the indicated dose was diluted in PBS to a final volume of 50 $\mu$l and added to the apical side of Transwells containing differentiated HBEC-ALI cultures for 2–3 h, then removed. At 72 h postinfection HBEC-ALI Transwells were washed once with PBS, dissociated with TrypLE, and fixed in 4% formaldehyde for 15 min. Fixed cells were washed and incubated for 15 min at room temperature in Perm/Wash buffer (#554723; BD). Permeabilised cells were pelleted, stained for 15 min at room temperature in 100 $\mu$l of sheep anti-SARS-CoV-2 nucleoprotein antibody (DA114; MRC-PPU) at a concentration of 0.7 $\mu$g/ml, washed, and incubated in 100 $\mu$l AF488 donkey anti-sheep (#713-545-147; Jackson ImmunoResearch) at a concentration of 2 $\mu$g/ml for 15 min at room temperature. Stained cells were pelleted and fluorescence staining analysed on a BD Fortessa flow cytometer. Fluorescent images of whole Transwell ALI cultures were captured using a Cellomics Arrayscan (VTI; Thermo Fisher Scientific). 121 fields of view per Transwell were captured at 10× magnification. The images were analysed using HCS Studio 2.0 Client Software.

### qRT-PCR

RNA was extracted using RNeasy Mini Kit (QIAGEN) according to the manufacturer's recommendation. Gene expression was quantified using SYBR Green I Dye (Life Technologies) on a QuantStudio 7 Flex Real-Time PCR System (Applied Biosystems). Data were analysed using $2^{-\Delta\Delta Ct}$ method and Design and Analysis Software Version 2.5, QuantStudio 6/7 Pro systems (Applied Biosystems). The following primers were used:

---

of B.1.1.7 SARS-CoV-2. There was a dose-dependent reduction in cellular infection with camostat pretreatment. Images of whole Transwell inserts were captured. Blue–Hoechst nuclear stain. Green–cells expressing SARS-CoV-2 viral nucleoprotein. Representative microscopy montages of the Transwell inserts after the indicated treatment. Scale bar: 500 $\mu$m. Experiments were performed using cells from Donors B1 and B2. **(D, E)** Quantitation of infection from (D), pretreatment with 2 × 15 min topical camostat. All SARS-CoV-2 nucleoprotein positive cells are quantified and presented as percentage of the total number of cells (DAPI) on each Transwell Each dot represents quantitation from a whole Transwell. Dots are coloured according to the donor. For bronchial cells, Transwells were infected on two separate occasions. For the nasal cells, this was a single large-scale experiment. For bronchial cells, $P$-values were calculated using a Kruskal–Wallis test with Dunn's correction for multiple comparisons (between both treatment conditions and the control). $P = 0.0120$ for 50 $\mu$M 2× topical treatment versus control. $P < 0.0001$ for 300 $\mu$M versus control. For nasal cells, $P$-values were calculated using a Mann–Whitney $U$ test, $P < 0.0001$. Representative microscopy images for nasal cells are shown in Fig S4.

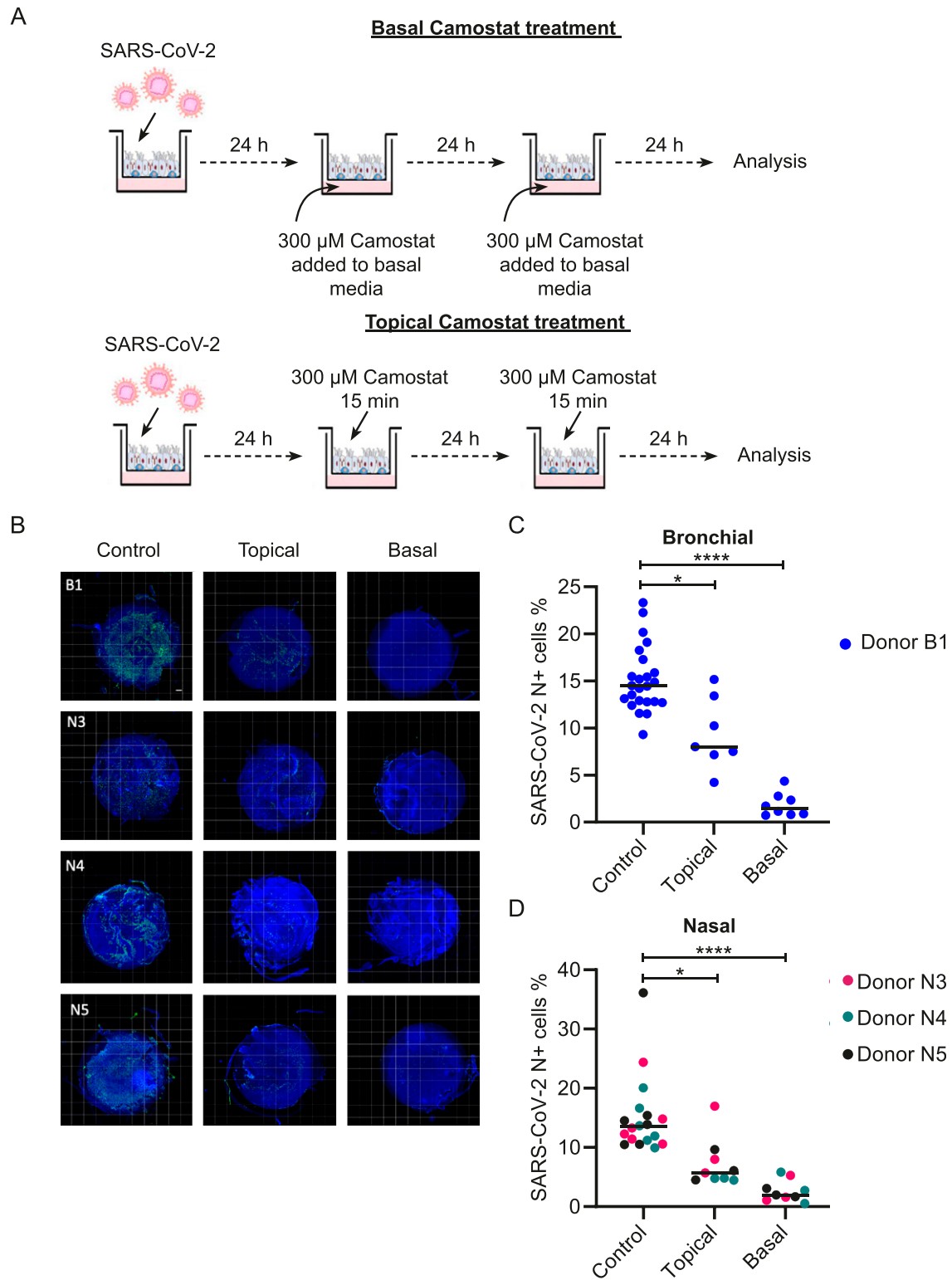

**Figure 4. Camostat attenuates cellular infection when commenced 24 h after exposure to SARS-CoV-2 virus. (A)** Schematic of experiment. A series of experiments was performed with B.1.1.7, this time with camostat administered after viral infection either for 48 h basally or topically for 15 min at 24 and 48 h after viral infection. **(B)** Images of whole Transwell inserts were captured using immunofluorescence. Blue–Hoechst nuclear stain. Green–cells expressing SARS-CoV-2 viral nucleoprotein. Representative microscopy montages of the Transwell inserts after the indicated treatment. Scale bar: 500 $\mu$m. Individual Transwells from four donors–B1, N3-5 are shown. In each case, both topical and basal camostat markedly reduced cellular infection. **(C, D)** Sustained basal exposure and 2 × 15 min topical exposure to camostat significantly reduced cellular infection in bronchial air–liquid interface cultures from Donor B1 and from nasal donors. Each dot represents quantitation from a whole

ACE2 Forward (5′-3′): CGAAGCCGAAGACCTGTTCTA; Reverse (5′-3′): GGGCAAGTGTGGACTGTTCC; TMPRSS2 Forward (5′-3′): CTGCTGGATTTCCGGGTG; Reverse (5′-3′) TTCTGAGGTCTTCCCTTTCTCCT; TBP Forward (5′-3′): AGTGAA-GAACAGTCCAGACTG; Reverse (5′-3′): CCAGGAAATAACTCTGGCTCAT.

## Histology

Transwells were washed three times with PBS, then fixed in 10% Neutral buffered formalin. The membrane was removed from the Transwell using a scalpel blade and paraffin embedded. 4-$\mu$m sections were cut and stained with haematoxylin and eosin.

## Immunofluorescence

Each Transwell was washed three times with PBS and fixed using 4% PFA for 15 min at room temperature before antibody staining. The following antibodies were used: anti-ACE2 antibody (Anti-ACE2 antibody—21115-1-AP; ProteinTech (Du et al, 2021); SARS-CoV/SARS-CoV-2 (COVID-19) spike antibody [1A9] (GTX632604; GeneTex); anti-SARS-CoV-2 nucleoprotein antibody (DA114; MRC-PPU); acetylated tubulin (T7451; Sigma-Aldrich); Muc5AC (MA5-12178; Invitrogen); Hoechst dye solution (100 $\mu$g/ml) was used for nuclei staining. Confocal images were taken using Nikon Confocal Microscopes C2, magnification 40× oil. Transwell inserts were imaged as described above.

## LDH cytotoxicity assay

For the positive and negative controls, 50 $\mu$l of 10× lysis buffer or ultrapure water, respectively, was applied to the apical surface of HBEC-ALI or nasal ALI Transwells and incubated at 37°C for 45 min. Then the basal media (500 $\mu$l) was used to wash the apical chamber. For each experimental condition, the basal media (500 $\mu$l) was used to wash the apical surface. Each condition was prepared in triplicate. Each sample was assessed for LDH release using the CyQUANT LDH Cytotoxicity Assay kit (C20300; Invitrogen) according to the manufacturer's recommendations. In brief, 50 $\mu$l of the final harvested sample was mixed with 50 $\mu$l of pre-prepared reaction mix and incubated for 30 min at room temperature. After stopping the reaction, plate absorbance was measured at 490 and 680 nm.

% Cytotoxicity was calculated as follows:

$$\%Cytotoxicity = \left[\frac{Compound-treated\ LDH\ activity - Spontaneous\ LDH\ activity}{Maximum\ LDH\ activity - Spontaneous\ LDH\ activity}\right] \times 100.$$

## Quantification and statistical analysis

Statistical analyses of mRNA expression assays and infection quantification data were performed using Prism 8 software (GraphPad Software). Statistical tests used are detailed in figure legends. $P$-values were noted as follows: ns, not significant; *$P < 0.05$; **$P < 0.01$; ***$P < 0.001$.

# Data Availability

Because of the nature of the data in this study, none has been deposited in a publicly available database. Request for further information on the data presented will be honoured by the corresponding authors.

# Supplementary Information

# Acknowledgements

SARS-CoV-2/human/Liverpool/REMRQ0001/2020 was a kind gift from Lance Turtle (University of Liverpool) and David Matthews and Andrew Davidson (University of Bristol). SARS-CoV-2 England/ATACCC 174/2020 was a kind gift from Greg Towers (University College London), and we are also grateful to Ajit Lalvani, Jake Dunning, Maria Zambon, and colleagues at Public Health England and Giada Mattiuzzo at the National Institute for Biological Standards and Controls and Wendy Barclay and Jonathan Brown and all colleagues in the United Kingdom Research and Innovation funded Genotype to Phenotype collaboration (G2PUK). Sheep anti-SARS-CoV-2 nucleoprotein antibody (DA114) was a kind gift from Paul Davies (obtained from the Medical Research Council Protein Phosphorylation Unit Reagents and Services, University of Dundee). LnCAP cells were a kind gift from Dr Charlie Massie. We gratefully acknowledge the support from Dr. Ravindra Mahadeva, Dr Jurgen Herre and Ms. Jacqui Galloway in establishing the primary cells from patients. We are grateful for the generous support of the United Kingdom Research and Innovation COVID Immunology Consortium (MR/V028448/1), Addenbrooke's Charitable Trust (15/20A) and the National Institute for Health Research (NIHR) Cambridge Biomedical Research Centre (BRC-1215-20014). This work was supported by a Wellcome Trust Principal Research Fellowship (084957/Z/08/Z) and Medical Research Council research grant MR/V011561/1 to PJ Lehner. This work was supported by the NC3Rs NC/S001204/1 project grant and the Roy Castle Lung Cancer Foundation grant (2015/10/McCaughan) to F McCaughan. This article presents independent research supported by the NIHR Cambridge Biomedical Research Centre (BRC). The NIHR Cambridge BRC is a partnership between Cambridge University Hospitals National Health Service Foundation Trust and the University of Cambridge, funded by the NIHR. The views expressed are those of the author(s) and not necessarily those of the NIHR or the Department of Health and Social Care.

## Author Contributions

W Guo: conceptualization, data curation, formal analysis, investigation, methodology, and writing—original draft, review, and editing.

Transwell from a single large-scale experiment. $P$-values were calculated using a Kruskal–Wallis test with Dunn's correction for multiple comparisons (between both treatment conditions and the control). $P = 0.0441$ and $P < 0.0001$ for Donor B1 50 and 300 $\mu$M versus control, respectively. $P = 0.0105$ for 300 $\mu$M topical versus control; <0.0001 for 300 $\mu$M basal versus control.

LM Porter: conceptualization, data curation, formal analysis, investigation, and methodology.

TWM Crozier: data curation, formal analysis, investigation, and methodology.

M Coates: resources and investigation.

A Jha: resources and investigation.

M McKie: data curation.

JA Nathan: conceptulisation, methodology, and writing—original draft.

PJ Lehner: conceptualization, resources, methodology, and writing—original draft.

EJD Greenwood: conceptualization, formal analysis, investigation, methodology, and writing—original draft, review, and editing.

F McCaughan: conceptualization, resources, formal analysis, supervision, funding acquisition, investigation, methodology, project administration, and writing—original draft, review, and editing.

## Conflict of Interest Statement

The authors declare that they have no conflict of interest.

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
