## [Reviewer comments · Life Science Alliance]

Life Science Alliance

Topical TMPRSS2 inhibition prevents SARS-CoV-2 infection in differentiated human airway cultures

Wenrui Guo, Linsey Porter, Thomas Crozier, Matthew Coates, Akhilesh Jha, Mikel McKie, James Nathan, Paul Lehner, Edward Greenwood, and Frank McCaughan

DOI: <https://doi.org/10.26508/lsa.202101116>

Corresponding author(s): Frank McCaughan, University of Cambridge and Edward Greenwood, University of Cambridge

Review Timeline:

Submission Date:	2021-05-09
Editorial Decision:	2021-07-07
Revision Received:	2021-10-20
Editorial Decision:	2021-11-11
Revision Received:	2022-01-04
Accepted:	2022-01-05

Scientific Editor: Novella Guidi

Transaction Report:

July 7, 2021

Re: Life Science Alliance manuscript #LSA-2021-01116-T

Dr. Frank McCaughan
University of Cambridge
Cambridge
UNITED KINGDOM

Dear Dr. McCaughan,

Thank you for submitting your manuscript entitled "Topical TMPRSS2 inhibition prevents SARS-CoV-2 infection in differentiated primary human airway cells" to Life Science Alliance. The manuscript was assessed by expert reviewers, whose comments are appended to this letter.

As you will note from the reviewers' comments below, Reviewer 3 is quite excited about these findings, but Reviewers 1 and 2 do have some concerns, with the common one being the unclarity of which cell source is used, the number of donors and number of independent experiments used. So please address this in the revised manuscript. Also, reviewer 2 is concerned that the protocols for basal and apical treatment are not consistent to draw such conclusions and therefore suggests authors to use the same concentrations and treatment times for apical and basal application and compare the efficacy of basal versus apical treatment at different time points post infection. Moreover, in DHT/enzalumatide treatment important controls are missing to support the conclusion. These concerns need therefore to be addressed by authors with additional experiments before resubmitting a revised version of the manuscript. All the other concerns raised by the reviewers should be addressed in the manuscript text without additional experimentation. We, thus, encourage you to submit a revised version of the manuscript back to LSA that responds to all of the reviewers' points.

The typical time frame for revisions is three months. Please note that papers are generally considered through only one revision cycle, so strong support from the referees on the revised version is needed for acceptance.

Thank you for this interesting contribution to Life Science Alliance. We are looking forward to receiving your revised manuscript.

Sincerely,

B. MANUSCRIPT ORGANIZATION AND FORMATTING:

Reviewer #1 (Comments to the Authors (Required)):

In this study, the authors used differentiated airway epithelial cells to study TMPRSS2 as a target for restricting infection of epithelial cells by SARS-CoV-2. The main new message is that apical treatment instead of basal treatment with the TMPRSS2 inhibitor camostat, already investigated in clinical trials for treatment of COVID-19, also blocks SARS-CoV-2 infection, and that this also applies to a variant of SARS-CoV-2. The authors used three sources of airway epithelial cells, but do not clarify sufficiently when which cell source is used, and also do not clarify the number of donors and number of independent experiments used. Furthermore, analysis of infection of the cells with SARS-CoV-2 is restricted to staining cells for viral proteins, assessed at a single time point after infection. No vRNA was assessed and no infectious viral particles were determined.

Specific comments:

1. Perhaps it is an oversimplification to state that only ciliated cells are targeted by SARS-CoV-2, since various publications also indicate that goblet cells and club cells, and cells expressing markers of goblet and ciliated cells may be targeted within the airways.
2. On page 5, please check whether references 11 and 21 are appropriate for the statement "we assessed TMPRSS2 regulation by androgens in primary airway cells".
3. Three sources of airway epithelial cells were used: Lonza, bronchial and nasal brushings. Both hAECs and HBEC cells are used as abbreviations for these cells and this is confusing. From the figures and figure legends it is not always clear which cells source is used for which experiment. Further characterization of the differentiation state of the various cultures using epithelial cell specific markers for basal, goblet, ciliated and club cells is needed.
4. Two different anti-ACE2 antibodies were used because the manufacturer could no longer provide the original one. Please describe how the new antibody was validated to obtain similar results as obtained with the old antibody, whether there were differences, and indicate which experiments were performed with which antibody.
5. The description of collection of apical washes for the LDH assay is not clear; also explain how 10 ul was recovered from the apical side, and whether this volume was sufficient to cover the apical surface of the cell layer, and whether the use of water instead of saline or PBES affected the outcome.
6. Figure 1: it is unclear which epithelial cells were used for figure 1, how many independent experiments were performed and how many different donors were used to generate the cultures. Therefore, the accuracy of the statistical analysis cannot be evaluated. The same comment also applies in part for the other figures.
7. Figure 1: can the authors exclude that the effect of enzalutamide is non-specific, since both TMPRSS2 and ACE2 are affected?
8. Figure 1: the only readout for SARS-CoV-2 infection used appears to have been staining for SARS-CoV-2 nuclear protein or S protein. Was no attempt made to assess vRNA levels and infectious viral particles by plaque assay? This outcome appears limited because perhaps inhibition of androgen signaling did not inhibit the % infected cells, but did affect viral titers. The same applies to Figure 2.
9. For figure 2F nasal and bronchial epithelial cells were used. Are these results derived from several independent experiments, using cells from separate donors? Please correct the unit for camostat used (now a distance instead of concentration).
10. Figure 3D, label y-axis: what is meant by objects?
11. Figure 3F: whereas camostat in Figure 2 was preincubated for 48 hours with the cells, in this figure only 15 minutes apical treatment was used. Please explain the difference, and indicate the volume applied to the apical side and the control treatment used.
12. Please explain why in some experiments S protein was used to identify infected cells, whereas in other experiment N protein is used.
13. In the discussion the authors indicate that camostat blocks sodium channel flux (secondary to inhibiting channel-activating proteases such as TMPRSS2). How can the authors be sure that the effect on SARS-CoV-2 infection can be explained only by

TMPRSS2 inhibition and not by an effect on sodium channel activity?

14. Page 18, last paragraph, second line: I guess "now:" should read "no". In the remainder of that line, it is unclear how the statement on monoclonal antibodies is linked to the unavailability of treatment options. Furthermore, trials using convalescent plasma are also worth mentioning here.

Reviewer #2 (Comments to the Authors (Required)):

The manuscript by Guo et al. investigated the efficacy of androgen receptor antagonists and the serine protease inhibitor camostat mesylate to inhibit SARS-CoV-2 entry and replication in primary human airway cells grown under air-liquid interface conditions. The authors observed that DHT and enzalutamide did not affect TMPRSS2 expression as well as SARS-CoV-2 replication in primary human bronchial and nasal cells. In contrast, camostat mesylate strongly reduced SARS-CoV-2 multiplication and spread in the cells. For camostat treatment the authors aimed to compare treatment of the airway cultures from the basal side and apical side (designated as topical application by the authors).

The novelty of the data is limited since several studies already investigated the efficacy of androgen receptor agonists on TMPRSS2 expression in human airway cells, although with controversial results. The antiviral efficacy of camostat mesylate against SARS-CoV-2 in human airway cell cultures has been demonstrated in a number of studies. Although the aim of comparing the antiviral activity of basal versus apical camostat treatment of airway cultures against SARS-CoV-2 is interesting, the data of the study do not support a final conclusion which application works better, since the protocols for basal and apical treatment (see below) are not consistent.

Most importantly, treatment of the cells with camostat from the apical side does not mimic topical treatment very well. Treatment of patients with aerosolized camostat mesylate requires formulation of the drug, since this is currently given oral. In order to investigate the antiviral activity of topical camostat it would be important to test such a formulated camostat preparation and even better in combination with a nebulizing chamber for airway cell cultures. To my opinion the antiviral activity of topical camostat mesylate against SARS-CoV-2 in human airway cells cannot be estimated from the present data.

Major comments:

1. DHT/enzalutamide treatment: Altered expression of TMPRSS2 in airway cells of male and female and upon treatment with AR antagonists has been investigated in several studies with controversial results. The data of the authors add on this studies, however, important controls are missing to support the conclusion. The authors observe no effect of DHT/enzalutamide on TMPRSS2 expression and SARS-CoV-2 replication. However, the authors should demonstrate that treatment of the cells worked e.g. by demonstrating altered expression of another AR target gene (e.g. FKBP5) in the cells as a positive control.
2. Camostat treatment from the basal chamber was performed for 48 h prior to infection - this is a very long incubation time. In contrast, apical treatment was performed for 15 min either 4 h prior to infection or 24 and 4h prior to infection. Due to the different incubation times and the dual treatment for the apical chamber it is difficult to compare the data. The authors should use the same concentrations and treatment times for apical and basal application for the experiments. In addition, all the treatments in the study are prophylactic. The authors should also compare the efficacy of basal versus apical treatment at different time points post infection. This has not been analysed in detail so far and would add on the novelty of the study.
3. The authors describe three different primary human airway cell cultures: hAEC, HBEC, nasal cells. However, it is not clear which cells are designated as hAEC - bronchial or nasal cells? Which cells were used for the experiments shown in figure 1 and figure 3? Have different cells been used for experiments shown in figure 2C (HBEC) and 2D (hAEC)?

Reviewer #3 (Comments to the Authors (Required)):

This is very interesting study on potential therapeutic strategies to prevent SARS-Cov2 viral entry to bronchial and nasal epithelial cells. It indicates that the androgen receptor signaling is not an effective target for these cells, whereas the topical use of camostat is a viable option.

The data are convincing and worth of further investigation in vivo.

Dear Dr Guidi,

Many thanks for sending on the reviews from our submission.

We are very grateful to the reviewers for the time and effort they have taken to review and improve our manuscript. Please find our response to their comments below:

Reviewer 1

Reviewer #1 (Comments to the Authors (Required)):

Thank-you. We are grateful to the reviewer for their comments.

In this study, the authors used differentiated airway epithelial cells to study TMPRSS2 as a target for restricting infection of epithelial cells by SARS-CoV-2. The main new message is that apical treatment instead of basal treatment with the TMPRSS2 inhibitor camostat, already investigated in clinical trials for treatment of COVID-19, also blocks SARS-CoV-2 infection, and that this also applies to a variant of SARS-CoV-2.

Thank-you for this summary.

The authors used three sources of airway epithelial cells, but do not clarify sufficiently when which cell source is used, and also do not clarify the number of donors and number of independent experiments used.

Thank-you. We have used multiple sources of airway cells and have amended the text to be clear as to which cells were used for each figure and experiment. Primary human airway cells are a challenging and significantly limited resource. A key strength of this study is that we have exclusively used primary cells and, where available, we have performed repeated experiments on primary cells from individual patients.

Furthermore, analysis of infection of the cells with SARS-CoV-2 is restricted to staining cells for viral proteins, assessed at a single time point after infection. No vRNA was assessed and no infectious viral particles were determined.

We chose to focus on immunofluorescence as the best measure of cellular infection.

Specific comments:

1. Perhaps it is an oversimplification to state that only ciliated cells are targeted by SARS-CoV-2, since various publications also indicate that goblet cells and club cells, and cells expressing markers of goblet and ciliated cells may be targeted within the airways.

We agree with this comment and have amended the text.

2. On page 5, please check whether references 11 and 21 are appropriate for the statement "we assessed TMPRSS2 regulation by androgens in primary airway cells".

Thank-you. These references were misplaced and not appropriate and we have amended the reference list.

3. Three sources of airway epithelial cells were used: Lonza, bronchial and nasal brushings. Both

hAECs and HBEC cells are used as abbreviations for these cells and this is confusing. From the figures and figure legends it is not always clear which cells source is used for which experiment. Further characterization of the differentiation state of the various cultures using epithelial cell specific markers for basal, goblet, ciliated and club cells is needed.

Thank-you. We have clarified for each figure and experiment the cells used. Figure 1b shows compelling differentiation and we have now included differentiation markers of bronchial and nasal ALI cultures as a Supplementary Figure.

4. Two different anti-ACE2 antibodies were used because the manufacturer could no longer provide the original one. Please describe how the new antibody was validated to obtain similar results as obtained with the old antibody, whether there were differences, and indicate which experiments were performed with which antibody.

Thank-you. We have amended figures to use images in which only one antibody (Proteintech) was used. It obtained similar results to the prior antibody (see Porter et al, Figure 1a - <https://www.biorxiv.org/content/10.1101/2021.09.08.459428v1.full>). It has also been fully validated in an independent publication - <https://www.nature.com/articles/s41467-021-25331-x#Sec10> .

5. The description of collection of apical washes for the LDH assay is not clear; also explain how 10 ul was recovered from the apical side, and whether this volume was sufficient to cover the apical surface of the cell layer, and whether the use of water instead of saline or PBES affected the outcome.

Thank-you for highlighting this. This section has been completely rewritten to add clarity. The volume used was 50µl.

6. Figure 1: it is unclear which epithelial cells were used for figure 1, how many independent experiments were performed and how many different donors were used to generate the cultures. Therefore, the accuracy of the statistical analysis cannot be evaluated. The same comment also applies in part for the other figures.

Thank-you and agreed. The cells used for many of the experiments, including all of Figure 1 were from Donor B1. All figure legends have been amended appropriately and the cells used clarified. In some panels cells from multiple donors have been used and this is now clearly presented.

7. Figure 1: can the authors exclude that the effect of enzalutamide is non-specific, since both TMPRSS2 and ACE2 are affected?

We have demonstrated that TMPRSS2 is regulated by both DHT and enzalutamide in HBECs from Donor B1 in a manner expected of an androgen-sensitive gene and mirroring the impact of DHT and enzalutamide on canonical androgen target genes. Please see response to Reviewer 2 point 1 also.

We agree that there is a reduction in ACE2 expression with the combination of DHT and enzalutamide and a trend towards reduction with enzalutamide alone. The presence or absence of an impact of enzalutamide on ACE2 is of potential interest but is beyond the scope of this article.

However, the key finding is that enzalutamide had the expected impact on the expression of TMPRSS2 but had no impact on cellular infection.

8. Figure 1: the only readout for SARS-CoV-2 infection used appears to have been staining for SARS-

CoV-2 nuclear protein or S protein. Was no attempt made to assess vRNA levels and infectious viral particles by plaque assay? This outcome appears limited because perhaps inhibition of androgen signaling did not inhibit the % infected cells, but did affect viral titers. The same applies to Figure 2.

As noted above we used cellular infection as denoted by expression of viral proteins only. We therefore cannot address any potential disconnect between the number of cells infected and viral titres. In general, in non-ALI cultures we would expect a consistent trend for both of these variables.

9. For figure 2F nasal and bronchial epithelial cells were used. Are these results derived from several independent experiments, using cells from separate donors? Please correct the unit for camostat used (now a distance instead of concentration).

Thank-you. Yes. We have now clarified both the data and the detail of the cells used in the legends. The legend has also been corrected.

10. Figure 3D, label y-axis: what is meant by objects?

Thanks and corrected.

11. Figure 3F: whereas camostat in Figure 2 was preincubated for 48 hours with the cells, in this figure only 15 minutes apical treatment was used. Please explain the difference, and indicate the volume applied to the apical side and the control treatment used.

Thanks. The purpose was not to compare basal and topical administration. Oral camostat is now known to be ineffective in humans in SARS-CoV-2 because of rapid metabolism in the circulation. The oral preparation is therefore not a viable therapeutic in COVID-19. However, the drug is effective if it can be delivered to the airway at the appropriate doses. Hence, we addressed whether a topical preparation could restrict SARS-CoV-2 infection.

Administration to the basal chamber is a mimic in vitro for systemic administration. This is a standard approach that others have used.

However the maximum systemic concentration after 600mg x4/day dosing in humans is <1 μ M (NCT04451083).

The basal experiments establish a robust proof of principle that TMPRSS2 inhibition is effective at adequate doses in our system. However, the systemic metabolism of camostat means concentrations >1 μ M could not be achieved in vivo by an oral preparation.

Topical airway dosing avoids systemic metabolism. Therefore, we asked whether a topical airway dose could be mimicked in vitro and proceeded to test doses that we knew from Phase 1 data could readily be administered nasally in clinical practice.

The duration of exposure was 15 mins as the nasal transit time is 20 minutes so a period of 15 minutes is achievable in nasal epithelium in human subjects.

A key point of this paper is that topical/apical TMPRSS2 inhibition is effective at restricting SARS-CoV-2 infection and that it has no obvious toxicity. We show that therapeutic concentrations are achievable after topical dosing. The 15min exposure period was because of the nasal transit time.

Importantly, much higher topical/nasal doses of camostat than were required to block SARS-CoV-2 infection in our experiments have been shown to be safe in humans with severe lung disease (Rowe et al, 2013) and the known pharmacology suggests that the covalent bond formed between camostat and TMPRSS2 will facilitate prolonged inhibition despite the brief nasal transit time (Breining et al, Coote et al)..

These points are now emphasised in the revised draft.

12. Please explain why in some experiments S protein was used to identify infected cells, whereas in other experiment N protein is used.

Immunofluorescence experiments were initially performed using an S-protein antibody as that was available first. When the nucleoprotein antibody became available it performed better – providing a stronger signal and was adopted for flow cytometry and later immunofluorescence experiments.

13. In the discussion the authors indicate that camostat blocks sodium channel flux (secondary to inhibiting channel-activating proteases such as TMPRSS2). How can the authors be sure that the effect on SARS-CoV-2 infection can be explained only by TMPRSS2 inhibition and not by an effect on sodium channel activity?

We cannot be sure as camostat does inhibit other serine proteases. However, there is a considerable body of data that suggests that TMPRSS2 is the critical cellular protease involved in spike cleavage and therefore extremely likely to be the key target for camostat in this context. We have amended the discussion to acknowledge this issue.

14. Page 18, last paragraph, second line: I guess "now:" should read "no". In the remainder of that line, it is unclear how the statement on monoclonal antibodies is linked to the unavailability of treatment options. Furthermore, trials using convalescent plasma are also worth mentioning here.

Thank-you. We have amended the discussion to reflect developments in the fast-moving field.

Reviewer #2 (Comments to the Authors (Required)):

Thank-you. We are grateful to the reviewer for their comments.

The manuscript by Guo et al. investigated the efficacy of androgen receptor antagonists and the serine protease inhibitor camostat mesylate to inhibit SARS-CoV-2 entry and replication in primary human airway cells grown under air-liquid interface conditions. The authors observed that DHT and enzalutamide did not affect TMPRSS2 expression as well as SARS-coV-2 replication in primary human bronchial and nasal cells. In contrast, camostat mesylate strongly reduced SARS-CoV-2 multiplication and spread in the cells.

Thank-you for this summary.

For camostat treatment the authors aimed to compare treatment of the airway cultures from the basal side and apical side (designated as topical application by the authors).

Thank-you. However, this was not a specific aim. We predicted (and have since been proven correct in multiple studies) that SARS-CoV-2 inhibitory concentrations of camostat cannot be attained after systemic (oral) delivery due to its rapid metabolism in the circulation. Therefore, we did not set out to compare the two approaches. We used the basal approach to show that TMPRSS2 inhibition could be effective in principle in our system. However basal administration mimics systemic dosing and we were aware it is not possible to achieve those concentrations in vivo after systemic dosing

Then, critically, we used the apical exposure for 15 minutes on two occasions to show that a clinically achievable exposure to this drug, after topical administration was sufficient to block cellular entry. The exposure time was chosen to approximate the nasal transit time of 20 minutes.

We apologise if the submitted draft was unclear and have emphasised these points in the revised manuscript.

The novelty of the data is limited since several studies already investigated the efficacy of androgen receptor agonists on TMPRSS2 expression in human airway cells, although with controversial results.

We agree that this is a controversial area and suggest that our data are both novel and add to the field.

The antiviral efficacy of camostat mesylate against SARS-CoV-2 in human airway cell cultures has been demonstrated in a number of studies.

We agree that a number of groups have demonstrated this as noted above and we have referred to a number of these studies.

Although the aim of comparing the antiviral activity of basal versus apical camostat treatment of airway cultures against SARS-CoV-2 is interesting, the data of the study do not support a final conclusion which application works better, since the protocols for basal and apical treatment (see below) are not consistent.

Thank-you. As noted above the aim was not to conclude that apical works better than basal.

Rather, we knew that basal (systemic) delivery is implausible in vivo due to drug metabolism after systemic delivery. This has now been borne out by the clinical trial data. The basal administration was a proof of principle that TMPRSS2 inhibition is effective in primary cells in our experimental system. However, sufficient camostat cannot be delivered to the airway cells via the oral/systemic route so the key question was whether it could be delivered apically or topically thus avoiding the metabolism in the circulation. The duration of epithelial cell contact for a nasal drug is 20 minutes (nasal transit time). Therefore, we chose to model prevention using two doses of 15 minutes which mimics two nasal sprays 24 hours apart. Of note long exposures to topical camostat is not appropriate since cells are being maintained in a physiologically relevant way at the air-liquid interface.

Most importantly, treatment of the cells with camostat from the apical side does not mimic topical treatment very well. Treatment of patients with aerosolized camostat mesylate requires formulation of the drug, since this is currently given oral. In order to investigate the antiviral activity of topical camostat it would be important to test such an formulated camostat preparation and even better in combination with a nebulizing chamber for airway cell cultures. To my opinion the antiviral activity of topical camostat mesylate against SARS-CoV-2 in human airway cells cannot be estimated from the present data.

Thank-you. However, we respectfully disagree. The formulation (dissolved in phosphate buffered saline) is almost identical to that used in the reference Rowe et al, 2013 (dissolved in saline) as a nasal spray in humans with a lung disease (cystic fibrosis). We did not test a nebulised formulation of the compound as we knew from Novartis that the large animal data (dogs) performed some years ago suggested minor inflammatory toxicity when nebulised to the lung. This stopped development of a lower airway inhaler / nebulised formulation being tested in humans despite the prior encouraging sheep data (Coote et al).

However, the dogs had no detectable toxicity with a nasal spray and Novartis moved to a Phase 1 clinical trial in Cystic Fibrosis patients.

We agree that there are certainly advantages to a nebulised system if we had wanted to mimic lower airway delivery but this is impractical in an academic class 3 laboratory with SARS-CoV-2 and we were aware of the dog data that precludes lower airway delivery in the clinic.

Finally, apical exposure to compounds in vitro has been shown to mimic drug delivery and toxicity in vivo - <https://doi.org/10.1093/toxsci/kfx255>

Major comments:

1. DHT/enzalumatide treatment: Altered expression of TMPRSS2 in airway cells of male and female and upon treatment with AR antagonists has been investigated in several studies with controversial results. The data of the authors add on this studies, however, important controls are missing to support the conclusion. The authors observe no effect of DHT/enzalumatide on TMPRSS2 expression and SARS-CoV-2 replication. However, the authors should demonstrate that treatment of the cells worked e.g. by demonstrating altered expression of another AR target gene (e.g. FKBP5) in the cells as a positive control.

Thank-you. We have added the suggested controls and are grateful for the suggestion. The data confirm that the androgen and enzalutamide altered the expression of canonical AR target genes.

2. Camostat treatment from the basal chamber was performed for 48 h prior to infection - this is a very long incubation time. In contrast, apical treatment was performed for 15 min either 4 h prior to infection or 24 and 4h prior to infection. Due to the different incubation times and the dual treatment for the apical chamber it is difficult to compare the data. The authors should use the same concentrations and treatment times for apical and basal application for the experiments. In addition, all the treatments in the study are prophylactic. The authors should also compare the efficacy of basal versus apical treatment at different time points post infection. This has not been analysed in detail so far and would add on the novelty of the study.

Thank-you. This point has been addressed in part above regarding the comparison of basal and apical dosing regimes. The basal regime was a proof of principle that targeting TMPRSS2 inhibition would be effective in our experimental system. The incubation time was less than in some prior studies (Suzuki et al – 5 days) and more than in others. We have explained above the logic for the duration of topical exposure.

We have now performed direct comparisons between basal and apical treatments using the B1.1.7 variant and for equivalent durations. The basal treatments are more effective – as would be predicted given the constant exposure. The key observation is that topical camostat is also effective at a concentration that is readily and safely achievable in humans.

We have performed intervention studies on bronchial and nasal airway cells from multiple donors. We thank the reviewer for the suggestion. These data suggest that camostat has activity when administered topically (for 15mins) 24h and 48h after infection. However, topical administration is less effective than continuous basal exposure; as would be predicted.

3. The authors describe three different primary human airway cell cultures: hAEC, HBEC, nasal cells. However, it is not clear which cells are designated as hAEC - bronchial or nasal cells? Which cells were used for the experiments shown in figure 1 and figure 3? Have different cells been used for experiments shown in figure 2C (HBEC) and 2D (hAEC)?

Thank-you. We are grateful for the question. We have now clarified the nomenclature and precisely which cells were used for each experiment.

Reviewer #3 (Comments to the Authors (Required)):

This is very interesting study on potential therapeutic strategies to prevent SARS-Cov2 viral entry to bronchial and nasal epithelial cells. It indicates that the androgen receptor signaling is not an effective target for these cells, whereas the topical use of camostat is a viable option. The data are convincing and worth of further investigation in vivo.

Thank-you. We are grateful to the reviewer for their comments.

November 11, 2021

RE: Life Science Alliance Manuscript #LSA-2021-01116-TR

Dr. Frank McCaughan
University of Cambridge
Medicine
Hills Road
Cambridge CB2 0JJ
United Kingdom

Dear Dr. McCaughan,

Thank you for submitting your revised manuscript entitled "Topical TMPRSS2 inhibition prevents SARS-CoV-2 infection in differentiated human airway cultures". We would be happy to publish your paper in Life Science Alliance pending final revisions necessary to meet our formatting guidelines.

- please upload your supplementary figures as single files also
- please add an Author Contributions section to your main manuscript text
- please consult our manuscript preparation guidelines <https://www.life-science-alliance.org/manuscript-prep> and make sure your manuscript sections are in the correct order
- please add callouts for Figures 4B-D; S2 and S3B to your main manuscript text
- please add Data Availability section

A. FINAL FILES:

B. MANUSCRIPT ORGANIZATION AND FORMATTING:

Sincerely,

Reviewer #1 (Comments to the Authors (Required)):

The authors have made extensive revisions based on the comments of the reviewers, and included post-infection experiments with camostat. However, the novelty of the findings, the limited analyses of infection (only immunofluorescence), the reliance on use of a single donor and mixing technical and biological replicates (two different donors) in the analysis for essential findings of the manuscript, are still major limitations. Nevertheless, these limitations are now more clearly described in the manuscript, so readers can judge these for themselves. Furthermore, I did not find a marked copy making it convenient to find changes to the manuscript (except for some isolated red-marked text).

Reviewer #2 (Comments to the Authors (Required)):

In the revised manuscript the authors provide a number of new data that have been requested by reviewer 1 and 2 in order to provide important control experiments or similar experimental conditions for e.g. apical versus basal treatment of airway cultures.

In my opinion apical treatment of differentiated airway cell cultures still does not mimic topical treatment of the respiratory epithelium very well, as already mentioned, and the explanation given by the authors does not fully convince me. However, the data in general are now solid for publication.

I have no further comments or suggestions.

January 5, 2022

RE: Life Science Alliance Manuscript #LSA-2021-01116-TRR

Dr. Frank McCaughan
University of Cambridge
Medicine
Hills Road
Cambridge CB2 0JJ
United Kingdom

Dear Dr. McCaughan,

Thank you for submitting your Research Article entitled "Topical TMPRSS2 inhibition prevents SARS-CoV-2 infection in differentiated human airway cultures". It is a pleasure to let you know that your manuscript is now accepted for publication in Life Science Alliance. Congratulations on this interesting work.

DISTRIBUTION OF MATERIALS:

Again, congratulations on a very nice paper. I hope you found the review process to be constructive and are pleased with how the manuscript was handled editorially. We look forward to future exciting submissions from your lab.

Sincerely,
